# Molecular Aspects and Treatment of Iron Deficiency in the Elderly

**DOI:** 10.3390/ijms21113821

**Published:** 2020-05-28

**Authors:** Antonino Davide Romano, Annalisa Paglia, Francesco Bellanti, Rosanna Villani, Moris Sangineto, Gianluigi Vendemiale, Gaetano Serviddio

**Affiliations:** Department of Internal Medicine, University of Foggia, 71121 Foggia, Italy; antonino.romano@unifg.it (A.D.R.); annalisapaglia@gmail.com (A.P.); francesco.bellanti@unifg.it (F.B.); rosanna.villani@unifg.it (R.V.); moris.sangineto@gmail.com (M.S.); gianluigi.vendemiale@unifg.it (G.V.)

**Keywords:** iron deficiency anemia, elderly, nutritional status

## Abstract

Iron deficiency (ID) is the most frequent nutritional deficiency in the whole population worldwide, and the second most common cause of anemia in the elderly. The prevalence of anemia is expecting to rise shortly, because of an ageing population. Even though WHO criteria define anemia as a hemoglobin serum concentration <12 g/dL in women and <13 g/dL in men, several authors propose different and specific cut-off values for the elderly. Anemia in aged subjects impacts health and quality of life, and it is associated with several negative outcomes, such as longer time of hospitalization and a higher risk of disability. Furthermore, it is an independent risk factor of increased morbidity and mortality. Even though iron deficiency anemia is a common disorder in older adults, it should be not considered as a normal ageing consequence, but a sign of underlying dysfunction. Relating to the molecular mechanism in Iron Deficiency Anemia (IDA), hepcidin has a key role in iron homeostasis. It downregulates the iron exporter ferroportin, inhibiting both iron absorption and release. IDA is frequently dependent on blood loss, especially caused by gastrointestinal lesions. Thus, a diagnostic algorithm for IDA should include invasive investigation such as endoscopic procedures. The treatment choice is influenced by the severity of anemia, underlying conditions, comorbidities, and the clinical state of the patient. Correction of anemia and iron supplementation should be associated with the treatment of the causal disease.

## 1. Introduction

The total iron body content of an adult is approximately 3–4 g, of which about 70% is contained both in the hemoglobin of the red blood cells and in the muscles as myoglobin. Iron is fundamental to many basic biological functions such as oxygen transport and the activities of many enzymes and cytochromes. Iron deficiency (ID) is the most frequent nutritional deficiency worldwide [1,2,3]. ID is the results of a persistent negative iron balance, in conditions of increased iron demand, and insufficient iron intake or increased iron loss. ID can lead to iron deficiency anemia. Up to 30% of anemia is related to ID, which is the second most common cause of anemia of the elderly [4]. According to WHO criteria, anemia is defined a blood hemoglobin (Hb) serum concentration <12 g/dL in women and <13 g/dL in men [5]. However, several authors suggest that a hemoglobin value lower than 12 g/dL is more appropriate in diagnosing anemia in both older men and women. Gender differences disappear with ageing, and Hb in the healthy elderly is generally lower than in younger people, decreasing during aging, even in the absence of clinical disorders [5,6]. Moreover, it seems that the risk of mortality and morbidity in older patients increases with Hb levels lower than 12.5 g/dL [7]. According to Röhrig et al., WHO hemoglobin reference values should be used for the elderly since anemia is not an aging-associated physiological condition [8].

The prevalence of anemia in people aged up to 65 years is about 17%, but increases to more than 20% in subjects aged over 85 years and can rise to 50% in chronically ill institutionalized patients. Anemia in the elderly is usually mild, with a mean Hb level >10 g/dL [9]. According to Stauder et al., 50% of male hospital inpatients and outpatients older than 80 years are affected by anemia [7]. A study by McLean et al. showed that ID prevalence ranges from 8.1% to 24.7% in community-dwelling elderly, 31% to 60% in nursing home older patients, and 40% to 72% in hospitalized elderly [10]. 

Since the percentage of elderly is expected to triple by 2050, anemia will become a significant clinical problem in the future. Definition of anemia in the elderly is controversial due to a lack of consensus on the diagnostic criteria, and this in turn reflects the high heterogeneity of the studied populations.

## 2. Signs, Symptoms, and Outcomes 

Anemia is frequently asymptomatic and is diagnosed during routine medical examinations in older adults. Progressive development of anemia in older people triggers adaptive mechanisms to preserve hemoglobin levels, so that signs and symptoms are usually present only in severe anemia. IDA is clinically characterized by decreased exercise tolerance, dyspnea, tachycardia, angina pectoris, paleness, edema, headache, restless leg syndrome, altered thermoregulation. These symptoms are caused by impairment of oxygen transport and compensatory mechanisms. In the elderly, anemia can lead to a deterioration of chronic clinical conditions, with the progression of frailty status and consequent reduction in physical performance and mobility, higher risk of falling, worsening of cognitive function, dementia, depression, delirium, and alteration of body composition with a decrease in bone and muscle density. These conditions characterize normal ageing and make the diagnosis of anemia very challenging. Specific IDA-related symptoms are related to a rapid turnover of epithelial cells which can lead to koilonychia, hair loss, stomatitis, and atrophic glossitis. Clinical signs related to specific causes (such as melena, hematemesis, hematuria, uterine bleeding, abdominal pain, diarrhea, weight loss) may be present even though these symptoms are less common in the elderly. It is important to point out that in the elderly, even mild anemia may lead to serious health consequences [2,3,11,12,13].

IDA in older adults negatively impacts on both health status and quality of life, and it is associated with a large number of negative outcomes such as longer hospitalization and a higher risk of disability. It is an independent risk factor of increased mortality and morbidity [10,11,12,13,14]. Moreover, IDA can precipitate clinical conditions in chronic diseases, such as heart and kidney failure, and chronic obstructive pulmonary disease [5,9]. 

## 3. Pathophysiology

### 3.1. Etiology

It is still unclear how physiological changes due to ageing may affect iron metabolism because available data are few and contrasting. At the moment, there is no demonstration that the increased incidence of IDA in the elderly is related to ageing physiological evolution [4]. The iron regulatory hormone hepcidin may play a role in the development of anemia in the elderly. Hepcidin, stimulated by inflammation cytokines, prevents iron release from macrophages and hepatocytes and inhibits the iron transport to plasma from the enterocytes of the proximal duodenum, increasing iron retention in the reticuloendothelial system [2,4,15]. Furthermore, there is an impaired erythropoietin production in response to anemia [7]. 

In 2018 the German Geriatric Society released two position papers on iron deficiency anemia in elderly, to place it in the wider context of geriatric syndromes. In the first paper, Röhrig et al. define IDA as a very common disorder in aged people, but one that was not caused by age-related physiological changes [8]. In the second paper, IDA is described as a geriatric syndrome, and thus similar to sarcopenia and frailty [16]. The complete absence of iron stores can be defined “absolute IDA”, whereas the terms “functional” or “relative” may indicate normal or higher iron stores with iron-restricted erythropoiesis, respectively. Even though IDA is a common disorder in older adults, it should not be considered as a normal ageing consequence, because it is usually a sign of an underlying disease. In the elderly, anemia is usually multifactorial (Figure 1). 

Definitive diagnosis of the cause of anemia in the elderly can be challenging, even if the main cause of anemia is identified in about 80% of cases. The prevalent cause of anemia in the geriatric population is ID. Nutrient deficiencies are implicated in one-third of anemia cases (and more than 50% of these are related to iron deficiency). Chronic diseases and chronic inflammations, are involved in about another one-third of anemia cases. Ageing is associated with a chronic low-grade inflammation and this has been named “Inflammaging” [17]. Inflammation in the elderly is associated with changes in body composition, immunosenescence, metabolic status and it is predictive of increased morbidity and mortality. The remaining 20–30% of anemia is defined as “unexplained anemia of the elderly” (UAE) or “idiopathic cytopenia of unknown significance” (ICUS), and is characterized by a hypoproliferative normocytic anemia in patients with no chronic diseases, inflammatory disorders, or nutritional deficiencies [4,5,6,9,11,18]. Gowanlok et al., studying a group of 570 patients older than 60 years, observed that erythropoietin levels were inappropriately low in anemia of unknown etiology, hypothesizing that decreased erythropoietin production may play a key role in the pathogenesis of this category of anemia [19,20]. The diagnosis of UEA may be considered after excluding other possible causes (including hematological disorders) [7]. In fact, the incidence of myelodysplastic syndrome in older people is high, rising to 36.4 cases/100.000 population per year in those over 80 years, and increasing further with ageing. The median age at diagnosis of myelodysplastic syndrome is 76 years. Long-term treatments with non-steroidal anti-inflammatory drugs, antithrombotic drugs, anticoagulants, proton pump inhibitors, corticosteroids, iron-free erythropoietin, may also be relevant causes of ID in the elderly [2,3,7,11].

Finally, low dietary iron intake could be a possible cause of ID. There are two types of dietary iron: heme form and non-heme form. Heme form is most prevalent in meat and meat products, oily fish, cereal products, eggs, pulses and dark green vegetable. Iron absorption is facilitated by vitamin C intake and prevented by coffee consumption. The evaluation of the effective amount of dietary iron intake is challenging in the elderly, as impaired adsorption and malnutrition are common [15,21].

### 3.2. Molecular Mechanisms

Systemic and cellular iron levels are tightly regulated. Iron is transported around the body by plasma transferrin and delivered to cells through an endocytotic process [22]. Within cells, iron is stored in ferritin, with hepatocytes and macrophages being particularly important sites of iron storage. Quantitatively, most iron is used for hematopoiesis, but all body cells have a requirement for iron. When demand is higher, iron is released from cells through the only known iron exporter, ferroportin [23]. Ferroportin is in turn controlled by hepcidin, a 25 amino acid peptide hormone secreted by the liver, which negatively regulates ferroportin by facilitating its internalization and degradation. As a result, iron accumulates within cells [24]. Thus, hepcidin plays a key role in iron homeostasis. As a playmaker of systemic iron balance, hepcidin transcription in the liver is fine controlled by multiple signals, especially erythropoietic drive, iron deficiency and inflammation [24]. In hepcidin-knock-out mice, iron accumulates in the liver, pancreas and heart, while plasma iron levels are reduced [25,26]. Overexpression of hepcidin in transgenic mice results in IDA with severe microcytemia [26]. Such observations have also been reported in humans. Patients with hepatic adenomas and anemia scarcely respond to iron replacement therapy, and have high levels of hepcidin expression and lower serum transferrin saturation [27]. When tissue and circulating iron levels are elevated, hepcidin transcription is upregulated to limit further iron entry, while hepcidin expression is downregulated with an increased erythropoietic drive. As noted above, in the elderly, low-grade inflammation is common. Acute inflammation contributes to the severity of anemia in the context of hospitalization. Multiple mechanisms are involved in the pathogenesis of anemia. Inflammatory cytokines such as TNF-α, IL-6, IL-β, IL-γ both slow erythropoiesis and increase hepcidin levels. IL-6 induces hepcidin transcription in response to multiple infections including streptococcus pneumonia and influenza A. IL-6-knockout mice demonstrated impaired or absent hepcidin induction in response to an inflammatory stimulus [28]. Such observations have important implications in the management of anemia in the elderly [29]. In these patients, a high level of IL-6 is expected to stimulate hepcidin expression which, in turn, induces iron retention in macrophages and reduces erythropoiesis [14]. Similar findings were demonstrated in dialyzed patients with end-stage renal disease [30].

## 4. Diagnosis

Accurate collection of a patient’s history (comorbidities, pharmacological therapy, symptoms), clinical evaluation, and laboratory tests are needed to perform a definitive and differential diagnosis of anemia. Laboratory tests should include serum Hb, complete blood count, mean cell volume (MCV), mean corpuscular hemoglobin (MCH), reticulocyte count, plasma ferritin, serum transferrin saturation, serum iron, serum folate and vitamin B12, serum copper, circulating inflammatory markers (such as C-reactive protein, and fibrinogen), and organ function markers as serum creatinine/eGFR, serum aminotransferases, serum electrophoresis, thyreotropin (TSH), lactate dehydrogenase (LDH), serum erythropoietin (EPO) [7,11]. Mild ID is characterized by a reduction of the transferrin saturation, with a normal Hb value. Severe ID is characterized by microcytic, hypochromic anemia [2,15].

Bone marrow puncture is the gold standard procedure to diagnose IDA [31]. This technique is crucial to diagnose eventual myelodysplasia, but it is not routinely used in aged and fragile patients because of its invasiveness. IDA is associated with microcytosis (MCV <80 fL). However, in the elderly, a vitamin B12 and folate deficiency may often occur, leading to an increase in MCV with resulting normocytic anemia that can complicate the interpretation of laboratory data. As a consequence, MCV assessment is not useful to rule out an IDA in the elderly, especially if patients present with comorbidities. Serum iron and transferrin saturation are the main indicators of adequate iron supply. The presence of low serum iron concentration (<33 g/dL) and transferrin saturation <15%, increased iron-binding capacity (>400 g/dL), and low serum ferritin value (≥50 μg/mL) is diagnostic for ID. However, the sensitivity of these parameters is lower in aged than in younger adult patients. In addition to being a marker of iron stores, plasma ferritin is an acute-phase marker, and may be increased in cases of systemic inflammation, infections, and chronic disorders. Thus, when serum markers of inflammation are higher, a serum ferritin lower than 100 μg/L can be strongly suggestive of ID. A serum ferritin lower than 12 μg/L indicates a lack of iron stores. Serum ferritin should be assessed in conjunction with other inflammatory markers (such as C-reactive protein, erythrocyte sedimentation rate, and fibrinogen), since its concentration can be elevated by inflammation and chronic disorders. An alternative marker of iron deficiency is the level of soluble transferrin receptor (sTfR) in serum. sTfR levels reflect erythropoietic activity and the demand of the bone marrow for iron. It increases in conditions of low iron concentration. Its use can be helpful to differentiate IDA from anemia caused by chronic diseases or inflammatory diseases, or by an interaction of both conditions. But its employment is limited because of the lack of standardized reagents in clinical laboratories, and the absence of a specific cut-off [2,5,11,32,33]. A situation of low serum iron associated with low transferrin saturation and high serum ferritin, with low total iron-binding capacity, is very suggestive of anemia of chronic diseases [1].

Reticulocyte count and reticulocyte index can help to differentiate between hypoproliferative anemia, and the presence of blood loss, hemolysis, hemoglobinopathies, or disorders in red cell structure and enzymes. In fact, the first condition is characterized by a reduction in reticulocytes, while the following is present with an increased red blood cells generation, because of bone marrow compensation. A reticulocyte index ≥2 is related to blood loss or hemolysis; if lower than 2, this indicates bone marrow failure, iron deficiency, or poor erythropoietin production. In elderly anemia, reticulocytes are usually low [1,9]. Mean reticulocyte hemoglobin content (CHr) and the reticulocyte hemoglobin equivalent (Ret-He) are indexes of the hemoglobin content in the reticulocytes. Low CHr or Ret-He values might be an early marker of iron-deficient erythropoiesis. They correlate with a lower iron supply for the red blood cell compartment in the bone marrow. Even though there are limited data about geriatric patients measurement of the Ret-He level is no more specific than other red cell indices (such as MCH and MCHC) in discriminating iron deficiency anemia from the anemia of chronic disease in the elderly [11].

The evaluation of serum hepcidin levels may prove a useful tool in distinguish IDA from the anemia associated with inflammatory and chronic diseases, but it requires further validation and is not yet routinely used in clinical practice.

In assessing IDA in the elderly, it is necessary to determine whether there are any underlying disorders that may cause iron deficiency. Chronic upper (40–60%) and lower (15–30%) gastrointestinal blood loss (as peptic ulcers, malignancies, esophagitis, varices in portal hypertension, gastritis, inflammatory bowel diseases, polyps, hemorrhoidal bleeding) are the most frequent conditions in older adults that could lead to IDA. Other causes are arteriovenous malformations (intestinal angiodysplasia is the most frequent one), extra-intestinal blood losses such as hematuria or gynecological bleeding, and other occult malignancies. Even though it is less frequent in industrialized countries, IDA could be related to nutritional deficiency, caused by malnutrition and insufficient iron intake, or by malabsorption disorders like celiac disease (CD) or achlorhydria. With particular regards to celiac disease, intestinal symptoms are less frequent in the elderly than in younger people. Moreover, micronutrient deficiencies may be prominent and sometimes the only presentation of the disease is in the elderly. CD is a cause of anemia, because of the malabsorption of iron, folic acid, and vitamin B12. Nevertheless, the impairment in iron absorption is the main driver of IDA in celiac elderly and often anemia is the only manifestation of the CD. Therefore, such observations reinforce the need for screening for CD in older patients with IDA [34,35].

Other conditions to consider in the differential diagnosis of IDA in the elderly are alcohol abuse, malnutrition, and chronic diseases such as chronic renal failure and inflammatory diseases [12,36,37,38,39].

Since blood loss commonly accounts for IDA in the elderly, the diagnostic algorithm should include endoscopic procedures, as esophagogastroduodenoscopy, colonoscopy, and possibly video capsule endoscopy. Indeed, 20% of elderly patients affected by IDA present with a negative upper and lower intestine endoscopy, and two-thirds of these patients suffer from a small bowel lesion. Thus, these investigations are recommended even with a negative occult fecal blood test result, because bleeding can be discontinuous. Old age is not an absolute contraindication to endoscopic procedures, but every patient needs to be evaluated, taking into consideration clinical conditions such as frailty and comorbidities. In selected cases, CT colonography (virtual colonoscopy) may be considered, even if its accuracy is lower than endoscopic investigations [7,11,12,38,40]. According to Andrès et al., iron deficiency in very old frail patients and in patients with life-threatening diseases should be treated with iron supplementation, avoiding preliminary invasive investigations [38].

## 5. Therapy

Treatment of IDA in the geriatric population should be based on the severity of anemia, taking into account comorbidities and the clinical state of each patient [1,7]. The aim of therapy is to correct anemia and replenish iron stores, in addition to treating the underlying disease. First-line therapy in adults is oral iron supplementation. The recommended daily dose ranges between 60–200 mg of elemental iron. The most common formulations include ferrous sulfate, gluconate, and fumarate. To date, there is no clear evidence of a different impact on hematological efficacy and side effects between all the available formulations. Nevertheless, divalent iron formulations such as ferrous sulfate and gluconate have shown a better bioavailability. Iron absorption is facilitated by an empty stomach, but epigastric pain and dyspepsia often develop after iron oral therapy. In these cases, patients prefer to take iron tablets with the main meals. Further possible adverse effects of iron oral therapy, occurring in 10–40% of patients, are diarrhea, constipation, nausea and vomiting, and melena. In addition to simultaneous food intake, changing the formulation of iron (ferrous gluconate tablets contain less iron), decreasing iron dose (the incidence of adverse effect is directly correlated to the dose amount), or using carbonyl iron may decrease the frequency of side effects and improve the patient’s compliance. Therapy failure is usually due to a lack of compliance (more than one in four patients stops taking their medications), mostly caused by adverse effects. Malabsorption could be another cause of treatment inefficacy. This latter condition may be diagnosed by the oral iron absorption test with liquid ferrous sulfate: in a fasting patient, the oral administration of 50–60 mg of iron should lead to an increase of serum iron of 100 μg/100mL in two hours. Particularly in the elderly, malabsorption can be caused by gastric hypochloridria secondary to atrophic gastritis, Helicobacter pylori gastric infection, and therapy with proton pump inhibitors. Therapy should be continued for at least 2–3 months after the correction of anemia, in order to replenish iron stores. A serum ferritin level higher than 100 μg/L can be considered as a therapy target. The need for long-term treatment is another important reason for discontinuing therapy in aged patients. In case of side effects, intolerance or inadequate compliance to oral therapy, poor gastrointestinal iron absorption, erythropoietin administration, severe iron deficiency, and chronic blood loss, intravenous iron replacement is indicated. The commonly employed formulations include iron sucrose, ferric carboxy-maltose, and iron dextran, with no evidence of better efficacy from one to another. In elderly patients, intravenous iron showed a higher efficacy and safety in different clinical trials, compared to oral treatment. Nevertheless, intravenous iron may cause adverse effects such as hypotension, arthralgias, myalgias, fever, cramps, and nausea. Side effects occur in 0.5% to 1% of treated patients. Anaphylaxis is the most dangerous side effect, occurring in one out of 200,000 patients. Intramuscular iron is not recommended because of the high risk of anaphylaxis and local side effects [1] (Table 1).

In 2017, a retrospective study evaluated the efficacy of parenteral iron administration in a non-hospitalized older population. Data interpretation was made difficult by the small number of aged people, receiving parenteral iron treatment. A significant increase in MCV, but a partial effect on hemoglobin and ferritin rates [41]. IDA in the elderly is usually a multifactorial disease, so that iron therapy could result ineffective. Treatment of other conditions may be required, such as vitamin B12 or folate deficiency, or by the resolution of secondary disorders and underlying diseases. Non-responder patients are usually affected by malignancies, renal failure, inflammatory diseases, persistent blood loss, or belong to the UAE group. Iron supplementation positively impacts chronic clinical conditions such as heart failure and obstructive pulmonary disease. Nevertheless, iron therapy in chronic diseases could promote the proliferation of microorganisms with consequential increased risk of infection [32,42]. In selected cases, such as hemodynamic instability or severe and symptomatic anemia, blood transfusion is necessary. Currently, there is no evidence of cut-off levels of Hb requesting blood transfusion in elderly, so the choice is dependent on the Hb levels correlated to the patient’s clinical conditions. A value of Hb 9–10 g/dL could be considered as a good target in older patients. However, each patient should be considered independently to assess the most appropriate treatment. Some have suggested a level of Hb <6 g/Ll to manage older patients in an intensive care unit performing blood transfusions. The American Association of Blood Bank recommends transfusion therapy in hemodynamically stable patients only if Hb levels are lower than 7 g/dL [43].

## 6. Conclusions

Iron deficiency is the most frequent cause of anemia in the elderly. Since aged patients are usually affected by multiple disorders, it is challenging to define if anemia status is caused by a lack of iron supply, an increase in iron losses, inflammatory disorders, chronic diseases, or by an interaction of all these factors. Cut-off ranges of traditional markers for iron status are not clearly defined in older people, and test results are sometimes difficult to interpret. Nevertheless, classical laboratory ranges are considered in many of the clinical trials and investigations. However, the underlying cause can be ruled out in almost 80% cases of anemia in the geriatric population. New laboratory parameters, such as hepcidin, are potentially useful for the diagnosis of IDA in the elderly, but their use is still limited. Searching for bleeding sources or malignancies is the first fundamental step after the diagnosis of IDA, since blood loss and cancer are two of the most common causes of IDA in the geriatric population. There are no specific guidelines for the treatment of IDA in the elderly, so that every single choice should be performed, taking into account the patient’s clinical conditions and their comorbidities. Treatment aims to replenish anemia and to supply iron stores. Higher Hb levels reduce the risk of frailty, morbidity and mortality, which are proportionally and independently related to the severity of anemia. The achievement of therapeutic targets could require longer treatment in older rather than younger people.

## Figures and Tables

**Figure 1 ijms-21-03821-f001:**
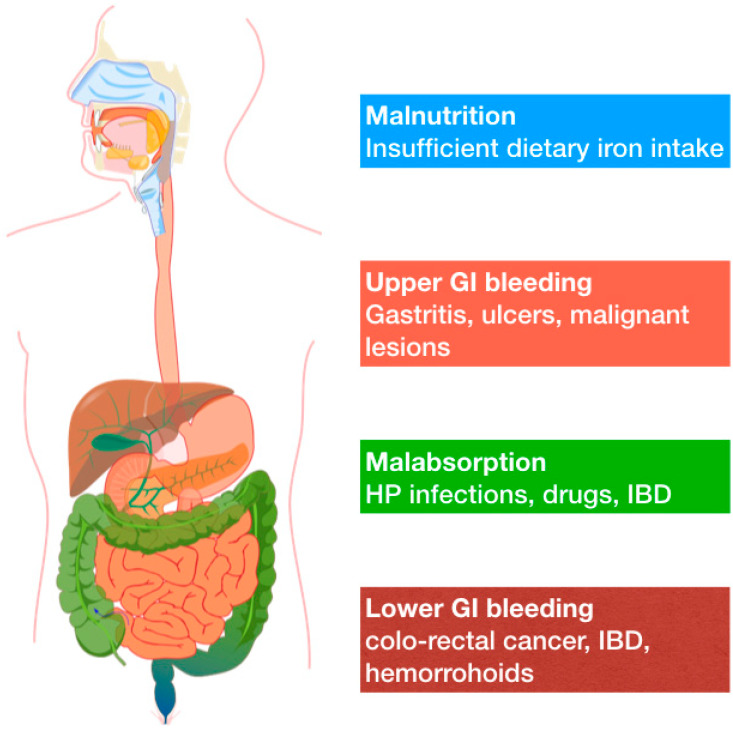
Common causes of IDA in adults include a broader range of potential mechanisms that are here synthetically depicted.

**Table 1 ijms-21-03821-t001:** Available oral and parenteral iron preparations with standard dosing. The list is not exhaustive; other oral and parenteral iron formulations may be available.

Drug	Elemental Iron Concentration	Dosing	
**IV IRON PRODUCTS**			
Ferric carboxymaltose	50 mg/mL	Weight ≥ 50 kg	2 doses of 20 mg/kg given every 7 days
		Weight ≤ 50 kg	2 doses of 15 mg/kg given every 7 days
Ferric gluconate	12.5 mg/mL	125 to 187.5 mg IV infusion over 1 h	
Iron sucrose	20 mg/mL	100 to 200 mg IV infusion over 15 min	
**ORAL IRON PRODUCTS**			
Ferrous gluconate	Various formulations	50 to 250 mg/day	
Ferrous sulfate	Various formulations	100 to 200 mg/day	
Iron sucrose	Various formulations	30 mg/day	

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
