# Peer review of "Molecular Aspects and Treatment of Iron Deficiency in the Elderly"

_ijms, 2020, doi:10.3390/ijms21113821_

Round 1
Reviewer 1 Report
This paper provides a good summary of data on iron deficiency anaemia in the aged. Almost half the amount of references are review papers. If possible data or statements mentioned should be taken from the paper publishing the original data and not from a review or text book.
As this paper is mainly based on clinical and epidemiological investigations, and the molecular data are not so prominent, the title may be modified into: "Molecular aspects and treatment of iron deficiency in the elderly".
I will not go into detail with respect to many mistakes in English language. Professional proofreading will be necessary. In particular many mistakes were seen in the lines 123-142. In lines 125-127, I saw even two times the same sentence.
Line 102. Add definition of "inflammaging" (chronic low-grade inflammation that develops with advanced age), reference Franceschi et al, Nature Reviews Endocrinology, 2018.
Almost all data on anemia and iron deficiency come from large scale epidemiological studies. The original publication, however, demonstrating that Hb parameters and iron metabolism are identical in prescreened healthy young adults and healthy aged (including double-isotope iron absorption investigation using a whole body counter). Mucosal uptake, mucosal transfer and retention of iron were even equally increased in both young and old subjects with iron deficiency. There was, however, a considerably lower red cell 59Fe-uptake in healthy young adults compared to healthy aged subjects. Ref. JJM Marx, Blood, 53:204-211, 1979.
Line 165-166, not bone marrow biopsy but bone marrow puncture was many years ago a golden standard for IDA. Biopsy is indeed needed for myelodysplasia.
Line 238, "replace" must be "repair".
Line 267, must be "one out of 200.000 patients".
Author Response
Reviewer 1
- ….the title may be modified into: "Molecular aspects and treatment of iron deficiency in the elderly".
Reply: modified as requested.
- Professional proofreading will be necessary. In particular many mistakes were seen in the lines 123-142. In lines 125-127, I saw even two times the same sentence.
Reply: the manuscript has been extensively revised by a native English speaker.
- Line 102. Add definition of "inflammaging" (chronic low-grade inflammation that develops with advanced age), reference Franceschi et al, Nature Reviews Endocrinology, 2018.
Reply: we comply with the reviewer’s request new sentences have been added (lines 102-105) with the reference indicated to better clarify the definition of inflammaging.
- Line 165-166, not bone marrow biopsy but bone marrow puncture was many years ago a golden standard for IDA. Biopsy is indeed needed for myelodysplasia.
Reply: modified as requested.
- Line 238, "replace" must be "repair".
Reply: modified as requested.
- Line 267, must be "one out of 200.000 patients".
Reply: modified as requested.
Reviewer 2 Report
Dear authors, I think the article is really good and it deals with a topic of great interest
However, I think it would be of interesting to go deeper into the information you provide on the relationship between celiac disease and IDA in the elderly.
The prevalence of CD in the elderly has significantly increased over the last two decades (Shiha et al., 2020) and remains underdiagnosed. Vitamin B12 deficiency is common in CD and may be responsible for anemia in the erderly. Jst as we have proposed in children (Martin-Masot et al., 2019),the possibility of association between both pathologies should always be taken into account. Please, discuss it in the present article
Author Response
Reviewer 2
We thank the reviewer for his revision and comments.
- …go deeper into the information you provide on the relationship between celiac disease and IDA in the elderly.
Reply: the point raised by the reviewer is very interesting and has been addressed in a new paragraph.
Round 2
Reviewer 1 Report
The manuscript is now acceptable for publication.
Professional in-house polishing of English language is still needed.
No additional revision of the content of this paper.
Author Response
Dear Reviewer,
thank you again for your valuable assistance in improving our paper. We further revised the English form as suggested.
Best regards.